# Preoperative Suffering of Patients with Central Neuropathic Pain and Their Expectations Prior to Motor Cortex Stimulation: A Qualitative Study

**DOI:** 10.3390/healthcare13151900

**Published:** 2025-08-04

**Authors:** Erkan Kurt, Richard Witkam, Robert van Dongen, Kris Vissers, Yvonne Engels, Dylan Henssen

**Affiliations:** 1Department of Neurosurgery, Unit of Functional Neurosurgery, Radboud University Medical Center, 6525 GA Nijmegen, The Netherlands; 2Department of Anesthesiology, Pain Medicine and Palliative Care, Radboud University Medical Center, 6525 GA Nijmegen, The Netherlands; jesper.witkam@radboudumc.nl (R.W.); robert.vandongen@radboudumc.nl (R.v.D.); kris.vissers@radboudumc.nl (K.V.); yvonne.engels@radboudumc.nl (Y.E.); 3Department of Medical Imaging, Radboud University Medical Center, 6525 GA Nijmegen, The Netherlands; dylan.henssen@radboudumc.nl

**Keywords:** central pain, motor cortex stimulation, qualitative study, patient expectations, chronic pain

## Abstract

**Objective**: This study aimed to improve the understanding of the lives of patients with chronic neuropathic pain planned for invasive motor cortex stimulation (iMCS) and assess their expectations towards this intervention and its impact. **Methods**: Semi-structured face-to-face interviews were conducted until saturation of data was reached. Patients were recruited from one university medical center in the Netherlands. All interviews were audio-recorded, transcribed verbatim, and subjected to thematic analysis using iterative and inductive coding by two researchers independently. **Results**: Fifteen patients were included (11 females; mean age 63 ± 9.4 yrs). Analysis of the coded interviews revealed seven themes: (1) the consequences of living with chronic neuropathic pain; (2) loss of autonomy and performing usual activities; (3) balancing energy and mood; (4) intimacy; (5) feeling understood and accepted; (6) meaning of life; and (7) the expectations of iMCS treatment. **Conclusions**: This is the first qualitative study that describes the suffering of patients with chronic neuropathic pain, and their expectations prior to invasive brain stimulation. Significant themes in the lives of patients with chronic pain have been brought to light. The findings strengthen communication between physicians, caregivers, and patients. **Practice Implications**: The insights gathered from the interviews create a structured framework for comprehending the values and expectations of patients living with central pain and reveal the impact of symptoms due to the central pain. This knowledge improves the communication between physicians and caregivers on one side and the patient on the other side. Furthermore, the framework enhances the capacity for shared decision-making, particularly in managing expectations related to iMCS.

## 1. Introduction

Invasive motor cortex stimulation (iMCS) is a form of neuromodulation to treat complex neuropathic pain syndromes and was first described by Tsubokawa et al. in the 1990s [1,2]. An electrode is placed epidurally over the primary cortex during an intracranial neurosurgical intervention [3]. Different electrode placement positions are described, both parallel and perpendicular to the primary motor cortex, resulting in different treatment results [4]. The use of multiple electrodes to increase the stimulation area has also been described and is associated with improved outcomes [5]. Neuro-imaging studies have shown iMCS-induced brain activity changes in the thalamus, anterior cingulate cortex, prefrontal cortex, and various regions in the brainstem [6]. However, the exact working mechanism of iMCS remains partially unknown. This poor understanding of the exact mechanisms of action is believed to explain why some patients do not respond to this complex treatment [7]. In a recent meta-analysis, it has been reported that 55–58% of patients experience a pain relief of >50% [8]. There are only limited qualitative studies focusing on the experiences of pain before treatment using a surgical procedure. Accardi-Ravid et al. used a qualitative descriptive approach to describe patient experiences of pain before and after surgery in a group of fourteen patients who underwent spine surgery [9]. They conclude that the results of their study can be used to guide future perioperative behavioral health services for patients undergoing spinal surgery and to establish realistic expectations of spine surgery. An integrative review [10] shows that spinal cord stimulation (SCS) positively affects different domains of life in patients with PSPS type 2, when assessed by a broad variety of outcome assessment instruments. The quantitative analyses suggest an overall improvement in most domains, although patients’ experiences show that limitations in daily life and living with the SCS system persist.

However, the effectiveness of pain treatments should not only be evaluated by measuring pain intensity scales. In fact, a multidimensional personalized approach is warranted [11]. This multidimensional evaluation of treatment has found its way into the evaluation of neuromodulation treatments as well [12]. A recent integrative review emphasized that, next to decreasing pain intensity scores, SCS in low back pain patients impacts a variety of other aspects, including quality of life and anxious and depressive thoughts [10]. Also, the expectations of patients [13,14] and the determination of goals [15] have been described as essential outcome parameters in neuromodulation treatment. However, little is known about the expectations and goals of chronic neuropathic pain patients who receive iMCS treatment. Therefore, this study investigated the expectations and goals of chronic neuropathic pain patients with regard to the iMCS treatment and outcomes. The insights from this study can be used to optimize patient education and to measure treatment effectiveness multidimensionally.

## 2. Methods

### 2.1. Study Design and Participants

A qualitative study was used to explore the patients’ expectations and goals with regard to the iMCS treatment and outcomes. Qualitative research holds the potential to uncover patterns that could not be elucidated through the use of quantitative research [16]. Semi-structured interviews were conducted to obtain nuanced descriptions and extensive, salient data of the patients’ perspectives. The interview topic list was based on the available literature [12,13,14]. The chosen topics have been suggested to be of importance when assessing patients with chronic neuropathic pain who undergo neuromodulation treatment. As there was no qualitative research on iMCS available when conducting this study, the researchers chose to study the available qualitative literature on spinal cord stimulation. Topics were discussed within the project team. Topics which were discussed during the interviews were (1) pain history of the patient; (2) the patient’s introduction to iMCS; (3) expectations of patients with regard to the iMCS procedure; (4) expectations of patients with regard to iMCS outcomes.

All patients were recruited by a neurosurgeon. Inclusion criteria comprised the following: (1) diagnosed with intractable central neuropathic pain (e.g., post-stroke pain, secondary trigeminal neuralgia); (2) pain duration for at least six months with a pain intensity scored on the numerical rating scale (NRS) score ≥5; (3) planned for iMCS; and (4) willing to participate in this interview and agree with the content of a standard informed consent form which states that their data can be used anonymously for research purposes such as writing and publishing a manuscript. This approval was then recorded in the electronic patient file. To seek in-depth information from a wide range of patients, purposive sampling was undertaken, meaning that both males and females with a range of ages and various daily activities were included. Patients without a full comprehension of the Dutch language were excluded from this study. All patients showed no motor or sensory aphasia and were able to do the interview without limitation in their vocabulary. The goal of this study was not to describe the impact of the intervention on the patient’s pain experience, but to recognize and describe themes that play an important role in the suffering of patients with central pain. We wanted to provide a framework for understanding the values and expectations of these patients.

### 2.2. The Process of Interviewing

Between 2019 and 2021, face-to-face semi-structured interviews were conducted by two researchers using the aforementioned topic list. Open-ended questions were used as starting points for the conversations. Patients were encouraged to express their own views freely and were regularly asked for clarification to ensure they were being understood correctly.

All interviews were conducted at the hospital within ten days prior to the iMCS operation, and the researchers ensured the patients that they were not involved in the iMCS procedure. Interviews lasted for a maximum of 45 min and were audio-recorded to be transcribed verbatim afterwards. As long as new codes emerged from the interviews, the inclusion of new participants continued. When no new codes were found, saturation appeared to be reached. In order to assure saturation, two additional interviews were conducted.

### 2.3. Data Analysis

Using the constant comparative method, the verbatim transcripts were analyzed by means of an inductive iterative process. Qualitative data analysis started after completion and transcription of the first interview. Codes derived from the previous interview were used as a starting point for subsequent coding. Additional codes were added when novel subjects emerged from the data. This coding process was carried out by two researchers independently using Atlas.ti software (v23.1.1; http://atlasti.com (accessed on 20 November 2023); ATLAS.ti Scientific Software Development GmbH, Berlin, Germany).

### 2.4. Ethical Approval

The study was performed according to the Good Clinical Practice guidelines. The Medical Review Ethics Committee concluded that this study was not subject to the Medical Research Involving Human Subjects Act (CMO Oost-Nederland; file number: 2016-3066).

## 3. Results

Out of fifteen patients invited for this study, all agreed to participate in the interview (comprising eleven females and four males). In retrospect, saturation of data was achieved after the thirteenth interview. The mean age of the participants was 63 years (standard deviation of 9.4 years). An overview of the included patients is provided in Table 1.

### 3.1. Themes Derived from the Interviews

Analysis of the coded interviews revealed seven themes: (1) the consequences of living with chronic neuropathic pain; (2) loss of autonomy and performing usual activities; (3) balancing energy and mood; (4) intimacy; (5) feeling understood and accepted; (6) meaning of life; and (7) the expectations of iMCS treatment (Figure 1).

This figure provides an overview of the themes that are derived from the interviews.

### 3.2. The Consequence of Living with Chronic Neuropathic Pain

All patients provided detailed descriptions of their pain history and previous procedures to manage their pain. It became evident that all patients had a long trajectory of visiting different physicians from diverse specialties. Various analgesic medications were reported to have a minor effect on the pain intensity, accompanied by significant side effects. These side effects significantly impacted their daily quality of life, resulting in discontinuing these medications or reducing the daily dosages. In daily life, patients experienced wide-ranging disturbances caused by their chronic neuropathic pain. On a so-called “bad day”, patients disclosed that they did not undertake any activity and stayed inside as much as possible. Three patients expressed an oversensitivity to sounds where regular sounds were experienced as loud noises. This also increased the pain sensations in two patients, which led them to spend less time with their grandchildren. Social life was further seriously impacted by living with chronic neuropathic pain. All patients disclosed that they regularly did not visit family gatherings, birthday parties, or other social events, as opposed to being frequent visitors before suffering from chronic neuropathic pain. Consequently, most patients expressed that their social circle decreased over time. This, in turn, had a significant impact on their social well-being and emotional well-being. This was described by most patients as a vicious circle from which they could not escape.

“*I haven’t been able to get much sleep, which leaves me with little energy. And then the pain, it also affects my mood. That, in turn, has consequences for my social life and for my partner as well*.”(Patient no. 7)

### 3.3. Loss of Autonomy and Performing Usual Activities

Most patients reported that they were no longer able to work or that they had to reduce working hours significantly. Consequently, they explained that they felt less autonomous in their lives. This also seemed to have a psychological impact on patients, especially on the former breadwinners. Three patients expressed that they were adamant about continuing their work during the first period of their illness, although they all needed to change their daily work activities. Next to the financial consequences, the loss of autonomy was also caused by the loss of social contacts at work.

“*I worked as a district nurse in a self-managing team, and there was always something to do. Then, I started working less and took on the scheduling. However, that soon became difficult as well, and I had to stop working because I was experiencing more and more side effects from medications. As a result, I lost my job, and now I’m on social benefits. I find that very challenging, and it makes me sad*.”(Patient no. 15)

Additionally, eleven patients disclosed that they often did not leave the house and, thereby, they felt a loss of autonomy. One patient expressed that she no longer visited supermarkets as she experienced this to be stressful. Other patients shared that they no longer take the bicycle to visit family members, friends, shops, or stores. To explain this, patients shared that they felt too exhausted to take the bicycle and, due to their orofacial pain, some also described that they suffered from excruciating pain when the wind touched their face while riding the bicycle. Since patients were no longer able to ride their bicycles as much, they expressed a severe decrease in their autonomy. The same was mentioned with regard to driving a car. Patients disclosed to be too exhausted or too overtaken by side-effects from their analgesic medications that they did not feel safe driving. Another frequently disclosed cause of loss of autonomy was the fact that they needed to ask others to help around the house with daily activities. Fourteen patients stated that their daily activities, mainly household chores, were significantly impaired by their pain. This made patients feel incapacitated.

“*On a better day, I do a little tidying around the house. I can manage dusting, but I can no longer vacuum. For that, I do have assistance*.”(Patient no. 5)

### 3.4. Balancing Energy and Mood

All patients expressed that their quality of sleep had significantly deteriorated due to their pain, and that they frequently felt exhausted. Three patients expressed taking naps during the day “to recharge”. One patient stated that his low energy levels were explained by the energy consumption of living in chronic pain. More than half of the patients detailed how their reduced sleep quality impacted everyday life. For example, social interactions, household chores, driving the car, and playing games and sports were all stated to be no longer possible when patients felt exhausted. It also impacted the feeling of being close to the partner, as some patients expressed sleeping separately from their partners. One patient explicitly expressed that he felt irritated by the thought that his wife was sleeping next to him whilst he was wide awake. This feeling increased his suffering. One patient expressed that this felt like a never-ending story because he no longer enjoyed relaxation and recreational activities due to his low energy levels caused by his sleeping deficits. This, in turn, meant that he no longer “recharged” while doing the things he loved and that he no longer had the energy to go on holiday with his wife. Moreover, he illustrated how his pain-induced deterioration of sleep quality impacted a significant part of his life. Therefore, feeling exhausted could also negatively impact patients’ moods. This negatively impacted mood was described as patients being irritable, having depressive thoughts, and feeling overly emotional.

“*My husband does mention that I’ve changed. I become easily irritated and, yes, somewhat snappy, I suppose. There are times when I respond very curtly or with a touch of grumpiness. Whereas naturally, I’m a very calm person*.”(Patient no. 1)

### 3.5. Intimacy

All patients disclosed that the chronic neuropathic pain significantly impacted their ability to be intimate with their partners. Eight patients reported that they lost their sexual drive, resulting in spending less time being intimate. Two patients explained that the loss of sensation in part of the body also had its impact on the genital region, resulting in the fact that sensations such as touch were experienced as less enjoyable or even painful. This was also explained by the six patients who suffered from orofacial pain. The painful sensation that was experienced when the facial region was touched or kissed resulted in couples being less intimate, not only in the bedroom, but in life in general. Three patients expressed that they did not necessarily miss this intimacy, although they knew that their partners did.

“*I do not even want to think about him touching my cheek or kissing my lips. It would be excruciatingly painful*.”(Patient no. 14)

### 3.6. Feeling Understood and Accepted

As previously described, most patients remarked that their social life had become much smaller since they were suffering from pain. This was explained to be at least partially caused by patients having the feeling that others did not understand their situation. Most patients expressed that they experienced a lack of support from people who stood nearby them. Although most friends and family members tried to advise the patients, this showed the patients how difficult it was for others to understand them. All patients expressed that at the beginning of their disease, they tried to explain their suffering to others. However, as they felt they were not understood, they stopped explaining over time.

“*Yes, there’s a lot of misunderstanding because it’s not visible. But if someone is interested, I’m willing to explain. However, I do expect them to remember and truly listen. The remarkable thing is that people whom you don’t know that well, or who happen to hear something about you and your pain, sometimes seem to understand it better and can also empathize more*.”(Patient no. 9)

### 3.7. Meaning of Life

All patients expressed that their chronic neuropathic pain severely impacted their sense of purpose and meaning in life. This loss was attributed to a combination of the themes mentioned above. Most notably, all patients reported a profound decrease in their overall joy in life. Two patients expressed that the only thing that got them through the week was the fact that they hoped that one day would be “a good one”. When such a day did not occur that week, it despaired those patients tremendously. Six patients expressed their desire to live for their partners, children, or grandchildren since these persons formed the remaining meaning of their lives. This was also a major driver for three patients to undergo iMCS treatment. All patients disclosed that the meaning of their lives had significantly decreased due to their condition.

“*I only keep going for my husband, my children and grandchildren. I really want to spend time with those little ones. That is one of the reasons why I want to try the iMCS treatment as well. If I did not have those people in my life, I would have stepped out earlier*.”(Patient no. 3)

### 3.8. The Expectations of iMCS Treatment

All patients reported that iMCS was still relatively unknown and that they felt they got this option by chance. Somewhere during their journey, the patients met a doctor who suggested the option of iMCS as something they had heard of once or twice. All participants explained that all referring doctors carefully explained that the procedure was experimental. Two patients believed that iMCS was going to be a panacea for their chronic neuropathic pain. In contrast, the other patients disclosed that they had learned from their previous pain treatments and understood that they should not have unrealistic expectations with regard to the effectiveness of iMCS. More specifically, with regard to the outcomes of iMCS, patients expressed a wide range of expectations. Most frequently, patients expressed that the pain reduction obtained using iMCS would have a significant impact on their social well-being and restoration of feeling autonomous. Nine patients reported that this pain reduction would allow them to go to social events more regularly, even if this resulted in more pain from time to time. It would be a price they were willing to pay. Also, patients hoped that their quality of sleep would improve after iMCS.

“*I hope that life will become more livable for me*.”(Patient no. 13)

## 4. Discussion and Conclusions

### 4.1. Discussion

This study is, to the authors’ knowledge, the first to provide qualitative insight into the lives of patients with chronic neuropathic pain who will undergo iMCS. The results suggest that the following themes are important for this patient category: (1) the consequences of living with chronic neuropathic pain; (2) loss of autonomy and performing usual activities; (3) balancing energy and mood; (4) intimacy; (5) feeling understood and accepted; (6) meaning of life; and (7) the expectations of iMCS treatment (Figure 1).

The outcomes further show that chronic pain causes a major impact on a wide spectrum of domains, impacting patients’ overall quality of life. It also emphasizes that chronic pain and the restrictions that come with it vary between patients. While some patients are primarily affected by the loss of autonomy, others are more deeply impacted by the erosion of intimacy (Figure 1). This variation underscores the importance of patient-centered care delivered by a multidisciplinary team in chronic pain management [17]. However, to optimize the outcomes of such a multidisciplinary and personalized approach, the identification of individual burden factors that shape the patient’s experience is a central component. The current manuscript offers a valuable framework that helps healthcare providers understand the areas in which patients need support, while also empowering patients to articulate the limitations they experience, ultimately enhancing the quality of care for patients with chronic pain. The burden and suffering of patients with central pain, as investigated in this study, may differ from those having other forms of chronic pain. This can be attributed to the fact that central pain patients not only endure chronic pain but also face additional neurological deficits. Nevertheless, certain themes may be relevant to all individuals living with chronic pain, regardless of the underlying cause. The themes derived from this study are overlapping results from other qualitative research in chronic neuropathic pain patients. For example, in SCS, similar themes have been derived by various groups [14,15,18]. Contrary to the study of Ryan et al., patients in this study seemed well-prepared with regard to iMCS, whereas Ryan et al. stressed the importance of better-designed, patient-friendly information materials with regard to SCS [19]. Patients expressed that they heard of iMCS almost “by chance” and they also stated that all referring physicians explicitly stressed the experimental nature of iMCS. Expectations of iMCS are believed to be greater when patients are told that the chances of success of the surgery are greater when they have responded to rTMS preoperatively as a predictor [20]. However, this did not have an impact on the present study outcomes, because rTMS was not provided to any of the included participants. This study emphasizes the importance of the use of qualitative methods concerning neuromodulation treatment regimens for chronic neuropathic pain patients. In the majority of publications on neuromodulation treatment of chronic neuropathic pain, outcomes are primarily assessed through the use of quantitative measures. However, it was underlined by a paper by Gjesdal and colleagues that pain relief with SCS is a complex and individual experience [21]. Furthermore, a recent letter to the editor underlined the importance of combining qualitative and quantitative outcome measures to obtain a complete overview of treatment effectiveness [22]. The proposed so-called holistic measure should be applicable in both clinical practice and research settings. Longitudinal follow-up with this multi-component endpoint might provide insights into long-term responsiveness to neuromodulation. In addition to this, it is important to obtain an overview of patients’ expectations prior to undergoing neuromodulation treatment, as Witkam et al. reported that patients’ expectations and experiences are variable [23]. They are not only variable, but may also vary over time in some patients. Therefore, it is important to define and discuss the pre-intervention expectations and goals in order not to have an unsatisfied patient later. This is confirmed by the study of Goudman et al. [15], as they stated that goal identification and proper documentation can be a valid method to obtain more personalized outcome measures after neuromodulation treatments [15]. In their study, a broad spectrum of individual patients’ goals, highlighting the need for individually targeted rehabilitation trajectories in the field of neuromodulation, was revealed. In our study, it appears that this group of patients suffering from central pain has realistic expectations from the iMCS therapy. However, not only are goals and expectations important to discuss, but limitations are also important to mention during preoperative counseling. Although most patients hoped to be able to participate more fully in activities, patients also experienced limitations such as the inability to do some physical activities due to the risk of falling, the unpredictability of some of the technology’s varying effects, and anxiety about the possibility of device failure. Participants were reluctant to travel too far from home in the case of an emergency [24].

Defining domains within qualitative research is challenging because QoL is diverse and shaped by subjective experiences. This may lead to overlap between themes. We chose not to merge individual themes under a broader category such as functional impairments, as this could diminish the personal nature of the experiences and potentially overlook themes explicitly mentioned by patients. After all, these were identified by patients as distinct themes for a reason. It is important to note that the themes were developed only after the interviews, based solely on the information derived from them.

In this study, assessments were primarily focused on qualitative outcome measures. It may be valuable to quantify the impact within each domain. Prospective studies may, therefore, want to use additional assessments (e.g., scale 0–10) to accurately quantify the changes within each qualitative domain (e.g., sleep quality, mood stability, pain intensity). This would help validate the qualitative findings obtained from the current study and would improve our understanding of the impact of motor cortex stimulation on patient outcomes. We recognize that further research with a more diverse and gender-balanced sample, including younger patients and males, is needed to explore how these themes might vary across different subpopulations.

### 4.2. Limitations

Although a previous study revealed that device-related inconveniences are an important factor in patients’ lives after neuromodulation implantation [25], the present study did not investigate the device-related inconveniences. Further, we cannot entirely rule out the influence of preoperative anxiety, but we believe its impact is minimal. This is due to the extensive preoperative counseling provided, as well as the fact that these patients already have substantial prior experience with anesthesia, surgeries, various pain treatments, and hospitalizations. Therefore, the effect of anxiety on the interview is expected to be limited. Another potential limitation of this study is that we relied on SCS literature to develop the interview guide. This was inevitable due to the absence of iMCS-specific research. While this might have introduced a theoretical limitation, input from the multidisciplinary team with experience in both SCS and iMCS helped tailor the topics to the iMCS context. Still, the findings should be considered exploratory, forming a starting point for future qualitative, iMCS-specific research. A restriction of this study is that no corrections were made for the duration of pain and the type of disease conditions in the study population. As a result, the correlation between the themes described and duration/type of condition could not be assessed. Finally, another limitation of our study could be the relatively small sample size in our study, although in our previous research in this field [14,23], we described that the saturation of data was reached in even smaller groups of patients. Others [26] confirm that data saturation can often be reached within 12 interviews, with key themes emerging even earlier. We recognize that studies involving greater variability, complexity, or specific subpopulations of chronic pain patients may require a larger number of interviews.

### 4.3. Conclusions

This is the first qualitative study on patients’ lives with chronic neuropathic pain prior to motor cortex stimulation and their expectations of its therapeutic effects. The findings underline the importance of understanding patients’ expectations in order to further improve iMCS, as our results suggest that pain intensity scores alone are not sufficient to monitor treatment outcomes. Important themes that play a role in the lives of patients suffering from chronic pain are revealed.

### 4.4. Practice Implications

Making the suffering of patients with chronic pain visible and identifying key themes in their daily lives that are important. As a result, their surroundings—including loved ones, family members, caregivers, and physicians—gain a clearer understanding of the factors that play a significant role. This increases awareness, fosters empathy, and recognition of the challenges faced by patients with chronic pain. Additionally, it highlights the expectations these patients have before undergoing brain stimulation treatment. As a result, the communication between physicians and caregivers on one side and the patient on the other side will improve.

## Figures and Tables

**Figure 1 healthcare-13-01900-f001:**
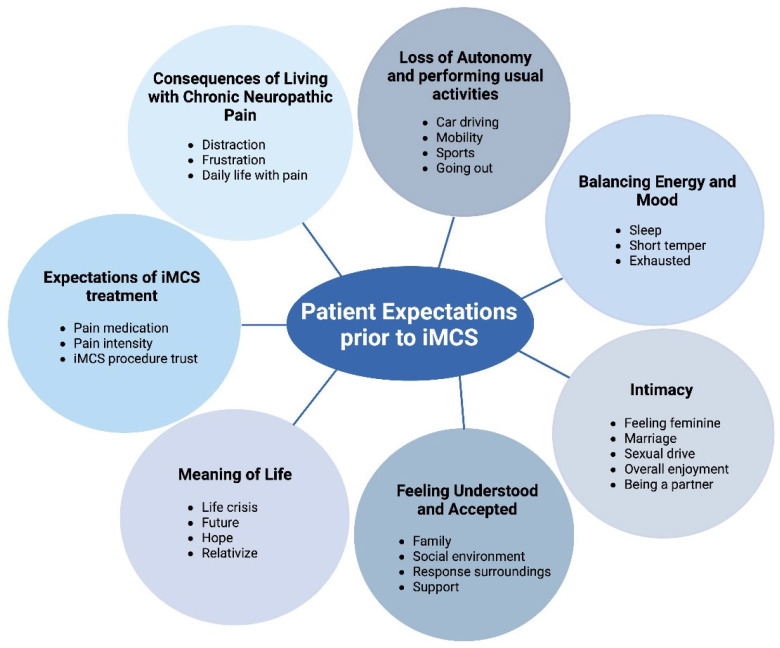
Overview of themes.

**Table 1 healthcare-13-01900-t001:** Overview of patients’ characteristics.

	Sex	Age	Pain Disorder	Side	Body Part	Duration of Pain (Years)	Mean Pain Intensity Prior to iMCS (NRS)
1	F	55	Wallenberg syndrome	Right	Hemi-face	4	8
2	M	61	Post-stroke pain	Right	Upper extremity	3	9
3	F	76	Trigeminal neuropathic pain	Left	Hemi-face	20	9
4	F	51	Postherpetic trigeminal neuralgia	Left	Hemi-face	4	9
5	F	62	Post-stroke pain	Left	Hemi-face	14	8
6	F	62	Post-stroke pain	Left	Hemi-body	6	8
7	F	52	Post-stroke pain	Left	Upper extremity	5	7
8	M	70	Post-stroke pain	Right	Hemi-face	7	8
9	F	59	Secondary trigeminal neuralgia	Left	Hemi-face	9	9
10	F	78	Post-stroke pain	Right	Hemi-body	3	8
11	F	72	Wallenberg syndrome	Left	Hemi-face	6	8
12	F	68	Post-stroke pain	Right	Hemi-body	3	8
13	F	69	Post-stroke pain	Left	Upper extremity	3	7
14	M	58	Trigeminal neuropathic pain	Left	Hemi-face	4	8
15	M	46	Trigeminal neuropathic pain	Right	Hemi-face	8	8

## Data Availability

Data are contained within the article.

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
