# Peer review of "Preoperative Suffering of Patients with Central Neuropathic Pain and Their Expectations Prior to Motor Cortex Stimulation: A Qualitative Study"

_healthcare, 2025, doi:10.3390/healthcare13151900_

Round 1

Reviewer 1 Report

Comments and Suggestions for Authors

Reviewer Comments on the Manuscript: “Preoperative Suffering of Patients with Central Neuropathic Pain and Their Expectations Prior to Motor Cortex Stimulation: A qualitative study”

General Comments:

Congratulations to the authors on this important and well-executed qualitative study. The integration of AI technology to analyze patient interviews is innovative and timely, especially in the context of chronic central pain syndrome, which is notoriously complex and multifactorial. While I am not a specialist in chronic pain management, I have observed that pain physicians often collaborate with various specialties to address the complex challenges their patients face. The effort to extract meaningful themes from patient narratives is commendable and offers valuable insights into the preoperative experiences and expectations of this patient population.

Major Comments:

The focus on developing qualitative questionnaires is highly relevant and useful for clinicians managing patients with chronic pain. As the authors rightly point out, chronic pain is influenced by a wide range of physiological, psychological, and emotional factors, making effective treatment a significant challenge.

The methodology used to derive themes from interviews using AI is particularly interesting and has the potential to be applied across various medical fields. This approach could enhance patient-centered care and improve satisfaction.

I wonder if the authors see some particular pattern in the themes described in Figure 1, depending on the duration of pain or the disease conditions. On a side note, the title or legends may be necessary in Figure 1. Furthermore, I wonder how the authors consider the themes described as specific to the patient groups chosen in this manuscript. I felt that these themes can apply to any candidate who suffers from pain for a long period of time.

In addition, I wonder how the framework described in this manuscript could provide additional benefits for patients with chronic pain, as a multidisciplinary team-based approach (including physicians, nurses, psychologists, etc.) is already recommended (https://www.anesthesiology.theclinics.com/article/S1932-2275(23)00024-1/fulltext). I would like to know what the authors expect to see by introducing this framework.

I understand it would not be necessary, but to further strengthen the manuscript, I would like the authors to consider including some quantitative data, if available. For instance, for the theme such as “balancing energy and mood,” it would be valuable to assess whether patients report improvements in sleep quality or mood post-treatment. A simple 0–10 scale could be used to quantify changes in specific domains (e.g., sleep quality, mood stability, pain intensity, etc.). This would help validate the qualitative findings and demonstrate the impact of motor cortex stimulation on patient outcomes, in this case.

Minor Comments:

Please ensure consistency in reference formatting—some citations appear before the period, while others appear after. Choose one style and apply it uniformly throughout the manuscript.

In the method, it would be better to indicate the number of patients who refused to participate in this interview, if applicable.

On line 58, please define the acronym SCS at first mention (presumably “spinal cord stimulation”) and use the abbreviation consistently thereafter.

On line 80, there appears to be a discrepancy in font type or size. Please adjust to maintain formatting consistency.

Author Response

Please see attached word document

Thanks, Erkan

Reviewer 2 Report

Comments and Suggestions for Authors

Major Concerns
Inadequate Sample Representativeness and Saturation Claims
The study includes only 15 patients (11 females), with a narrow age range (mean 63 years) and specific pain etiologies (predominantly post-stroke and trigeminal neuropathic pain). The authors claim data saturation was reached after 13 interviews, but the small sample size raises doubts about whether the themes adequately capture the diversity of experiences in central neuropathic pain populations. For instance, the lack of male participants and younger patients limits generalizability, and the homogeneous clinical profiles may overlook unique challenges faced by broader patient groups.

Methodological Gaps in Thematic Analysis
While the study employs inductive coding, the description of the analytical process is insufficient. There is no transparency regarding inter-coder reliability (e.g., how disagreements were resolved or Cohen’s kappa values), nor does the report detail how codes were refined across iterations. The reliance on spinal cord stimulation (SCS) literature to design interview topics for motor cortex stimulation (iMCS) patients introduces a theoretical mismatch, as iMCS-specific expectations and experiences may differ fundamentally from SCS.

Overlapping and Undefined Thematic Categories
The seven identified themes show significant overlap (e.g., "Loss of Autonomy" and "Balancing Energy and Mood" both relate to functional impairment). This suggests a lack of rigorous thematic refinement, potentially leading to superficial categorization rather than deep theoretical insight. For example, the theme "Meaning of Life" is cursorily linked to family relationships without exploring existential struggles in depth, which weakens the interpretive depth.

Incomplete Discussion of Limitations
The authors fail to address critical limitations, such as the short interview window (10 days preoperatively), which may introduce bias from acute preoperative anxiety. Additionally, the study does not acknowledge how patients’ prior treatment failures (e.g., medication side effects) might color their expectations of iMCS, potentially skewing data toward hopefulness or skepticism. The exclusion of non-Dutch speakers further restricts cultural diversity.

Minor Concerns
Inconsistent Terminology and Reporting
The abstract states "central neuropathic pain," but the results focus primarily on post-stroke and trigeminal pain, which may not fully represent central pain etiologies (e.g., multiple sclerosis-related pain). Additionally, the NRS pain scores (mean 8) are reported without contextualizing how patients defined "bad days," limiting interpretation of pain impact.

Superficial Link to Existing Literature
The discussion cites prior qualitative studies on SCS but does not sufficiently differentiate iMCS’s unique surgical and neurophysiological mechanisms. For example, the experimental nature of iMCS is mentioned, but the study does not explore how patients reconcile this with their treatment expectations, missing an opportunity to analyze cognitive dissonance or informed consent processes.

Recommendations for Revision
1. Expand the sample to include diverse patient demographics (e.g., male patients, younger individuals, varied pain etiologies) and provide quantitative measures of saturation (e.g., thematic density plots).
2. Report inter-coder reliability statistics and detail the iterative coding process, including examples of code refinement.
3. Revise thematic categories to eliminate overlap and deepen theoretical grounding, perhaps through integration of social suffering or disability theory.
4. Address preoperative anxiety as a potential confounder and explore how prior treatment experiences shape iMCS expectations.
5. Strengthen the discussion by explicitly linking iMCS-specific factors (e.g., invasive surgery risks, electrode placement variability) to patient narratives.

The study provides a preliminary exploration of patient experiences but suffers from methodological weaknesses and interpretive shallowness. Without addressing the above concerns, the findings lack the rigor needed to inform clinical practice or theoretical development in neuromodulation for central pain.

Comments on the Quality of English Language

Major Concerns
Inadequate Sample Representativeness and Saturation Claims
The study includes only 15 patients (11 females), with a narrow age range (mean 63 years) and specific pain etiologies (predominantly post-stroke and trigeminal neuropathic pain). The authors claim data saturation was reached after 13 interviews, but the small sample size raises doubts about whether the themes adequately capture the diversity of experiences in central neuropathic pain populations. For instance, the lack of male participants and younger patients limits generalizability, and the homogeneous clinical profiles may overlook unique challenges faced by broader patient groups.

Methodological Gaps in Thematic Analysis
While the study employs inductive coding, the description of the analytical process is insufficient. There is no transparency regarding inter-coder reliability (e.g., how disagreements were resolved or Cohen’s kappa values), nor does the report detail how codes were refined across iterations. The reliance on spinal cord stimulation (SCS) literature to design interview topics for motor cortex stimulation (iMCS) patients introduces a theoretical mismatch, as iMCS-specific expectations and experiences may differ fundamentally from SCS.

Overlapping and Undefined Thematic Categories
The seven identified themes show significant overlap (e.g., "Loss of Autonomy" and "Balancing Energy and Mood" both relate to functional impairment). This suggests a lack of rigorous thematic refinement, potentially leading to superficial categorization rather than deep theoretical insight. For example, the theme "Meaning of Life" is cursorily linked to family relationships without exploring existential struggles in depth, which weakens the interpretive depth.

Incomplete Discussion of Limitations
The authors fail to address critical limitations, such as the short interview window (10 days preoperatively), which may introduce bias from acute preoperative anxiety. Additionally, the study does not acknowledge how patients’ prior treatment failures (e.g., medication side effects) might color their expectations of iMCS, potentially skewing data toward hopefulness or skepticism. The exclusion of non-Dutch speakers further restricts cultural diversity.

Minor Concerns
Inconsistent Terminology and Reporting
The abstract states "central neuropathic pain," but the results focus primarily on post-stroke and trigeminal pain, which may not fully represent central pain etiologies (e.g., multiple sclerosis-related pain). Additionally, the NRS pain scores (mean 8) are reported without contextualizing how patients defined "bad days," limiting interpretation of pain impact.

Superficial Link to Existing Literature
The discussion cites prior qualitative studies on SCS but does not sufficiently differentiate iMCS’s unique surgical and neurophysiological mechanisms. For example, the experimental nature of iMCS is mentioned, but the study does not explore how patients reconcile this with their treatment expectations, missing an opportunity to analyze cognitive dissonance or informed consent processes.

Recommendations for Revision
1. Expand the sample to include diverse patient demographics (e.g., male patients, younger individuals, varied pain etiologies) and provide quantitative measures of saturation (e.g., thematic density plots).
2. Report inter-coder reliability statistics and detail the iterative coding process, including examples of code refinement.
3. Revise thematic categories to eliminate overlap and deepen theoretical grounding, perhaps through integration of social suffering or disability theory.
4. Address preoperative anxiety as a potential confounder and explore how prior treatment experiences shape iMCS expectations.
5. Strengthen the discussion by explicitly linking iMCS-specific factors (e.g., invasive surgery risks, electrode placement variability) to patient narratives.

The study provides a preliminary exploration of patient experiences but suffers from methodological weaknesses and interpretive shallowness. Without addressing the above concerns, the findings lack the rigor needed to inform clinical practice or theoretical development in neuromodulation for central pain.

Author Response

please see attached word document

thanks, Erkan

Round 2

Reviewer 2 Report

Comments and Suggestions for Authors This is a well-conducted and insightful qualitative study that makes a valuable contribution to the field of chronic neuropathic pain management, particularly regarding patients awaiting invasive motor cortex stimulation (iMCS). The strength of this research lies in its focus on a previously underexplored area: the preoperative experiences and expectations of patients with central neuropathic pain preparing for iMCS. By using semi-structured interviews and thematic analysis, the authors have successfully uncovered seven meaningful themes that vividly capture the multifaceted impact of chronic pain on patients' lives. It's impressive to see how the study goes beyond mere pain intensity to explore crucial aspects like loss of autonomy, changes in intimacy, and shifts in life meaning—elements that truly reflect the patient perspective. The methodological approach is rigorous, with clear steps outlined for participant recruitment, data collection, and analysis. The achievement of data saturation and the use of independent coding by two researchers enhance the credibility of the findings. Including direct patient quotes throughout the results section effectively illustrates the themes and adds a genuine, human touch to the analysis. The identified themes provide important insights for clinical practice, particularly in improving physician-patient communication and supporting shared decision-making. It's valuable that the study highlights the need for a personalized, multidimensional approach to assessing treatment outcomes, moving beyond traditional pain scales. Overall, this is a significant piece of research that fills a gap in the literature. The findings are not only academically relevant but also practically useful for healthcare professionals involved in the care of patients with chronic neuropathic pain. I strongly support the publication of this study, as it enriches our understanding of patient experiences and can inform better clinical practices in pain management.

Author Response

Thank you for your meticulous reading of our manuscript. We believe the comments you and reviewer 1 provided indeed improved our manuscript significantly, many thanks